A third-generation dispersion and third-generation hydrogen bonding corrected PM6 method: PM6-D3H+

Kromann Jimmy C. 1
Christensen Anders S. 1
Steinmann Casper 2
Korth Martin 3
Jensen Jan H. 1 jhjensen@chem.ku.dk
1 Department of Chemistry, University of Copenhagen , Denmark
2 Department of Physics, Chemistry and Pharmacy, University of Southern Denmark , Denmark
3 Institute of Theoretical Chemistry, Ulm University , Germany
Vondrasek Jiri
Electronic publication date: 2014 Jun 19
Publication date: 2014
Volume: 2
Electronic Location ID: e449
Received 2014 Apr 4; Accepted 2014 Jun 4
Copyright: © 2014 Kromann et al.
Copyright year: 2014
Copyright holder: Kromann et al.
License: This is an open access article distributed under the terms of the Creative Commons Attribution License, which permits unrestricted use, distribution, reproduction and adaptation in any medium and for any purpose provided that it is properly attributed. For attribution, the original author(s), title, publication source (PeerJ) and either DOI or URL of the article must be cited.
License URL: https://creativecommons.org/licenses/by/4.0/

Keywords: Computational chemistry, Proteins, Biochemistry, Computational biochemistry, Molecular modeling

Funding: Barbara Mez-Starck foundation MK received funding from the Barbara Mez-Starck foundation. The funders had no role in study design, data collection and analysis, decision to publish, or preparation of the manuscript.

==============================
We present new dispersion and hydrogen bond corrections to the PM6 method, PM6-D3H+, and its implementation in the GAMESS program. The method combines the DFT-D3 dispersion correction by Grimme et al. with a modified version of the H+ hydrogen bond correction by Korth. Overall, the interaction energy of PM6-D3H+ is very similar to PM6-DH2 and PM6-DH+, with RMSD and MAD values within 0.02 kcal/mol of one another. The main difference is that the geometry optimizations of 88 complexes result in 82, 6, 0, and 0 geometries with 0, 1, 2, and 3 or more imaginary frequencies using PM6-D3H+ implemented in GAMESS, while the corresponding numbers for PM6-DH+ implemented in MOPAC are 54, 17, 15, and 2. The PM6-D3H+ method as implemented in GAMESS offers an attractive alternative to PM6-DH+ in MOPAC in cases where the LBFGS optimizer must be used and a vibrational analysis is needed, e.g., when computing vibrational free energies. While the GAMESS implementation is up to 10 times slower for geometry optimizations of proteins in bulk solvent, compared to MOPAC, it is sufficiently fast to make geometry optimizations of small proteins practically feasible.

Introduction

Dispersion and hydrogen bonded corrections to the PM6 method (Stewart, 2007) such as PM6-DH2 (Korth et al., 2010), PM6-D3H4 (Řezáč & Hobza, 2012) and PM6-DH+ (Korth, 2010) yield interaction energies that in many cases rival in accuracy those computed with Density Functional Theory (DFT) (Yilmazer & Korth, 2013; Korth & Thiel, 2011). The computational efficiency of the underlying PM6 method allows for calculations that are not practically possible with DFT or Hatree–Fock (HF), such as geometry optimizations of proteins or vibrational analyses of large systems. For example, recent studies by Gilson (Muddana & Gilson, 2012) and Grimme (2012) have used dispersion and hydrogen bonded PM6 (PM6-DH+ and PM6-D3H respectively) to compute the vibrational free energy contribution to the standard binding free energy for host–guest systems and have demonstrated that the added contributions make a crucial contribution.

However, computing this vibrational free energy contribution can be complicated by the presence of one or more imaginary frequencies in the vibrational analysis (H Muddana and MK Gilson, pers. comm., 2013). The source of these imaginary frequencies are usually numerical errors amplified by a flat potential energy surface and the imaginary frequencies often correspond to low lying frequencies that make a significant contribution to the vibrational entropy. Thus, these numerical problems can introduce a significant error in the binding free energy.

Preliminary calculations suggested that one of the sources of the imaginary frequencies in PM6-DH+ calculations using MOPAC could be solved by using different geometry optimization algorithms. To test this we implemented a new variant of PM6-DH+, called PM6-D3H+, in the GAMESS program (Schmidt et al., 1993) to allow us to test the use of the optimization algorithms implemented therein. PM6-D3H+ differs from PM6-DH+ in that the dispersion term is the third generation dispersion model developed by Grimme et al. (2010) rather than the Jurecka-type model developed by Jurečka et al. (2007). In that respect, PM6-D3H+ is identical to the PM6-D3H model developed by Grimme (2012) which has not yet been incorporated into a quantum chemistry program. This dispersion model was mainly chosen for convenience (as it was already implemented in GAMESS) and has little effect on the average accuracy compared to PM6-DH+ (although the maximum errors observed for the training set decrease). However, we show that PM6-D3H+ implemented in GAMESS results in vibrational analyses with significantly fewer imaginary frequencies than PM6-DH+ implemented in MOPAC (Stewart, 2012; Maia et al., 2012), due mainly to differences in geometry optimization algorithms and convergence criteria.

Theory

The energy model PM6-D3H+ has three contributions (1) EPM6-D3H+=EPM6+ED3+EH+.

Here, each contribution is a standalone semi-empirical module in GAMESS. These are discussed below.

PM6 implementation in GAMESS

E(PM6) is the molecular PM6 (Stewart, 2007) energy, which is taken as the gas phase energy unless otherwise noted. As part of this work we implemented the PM6 method in the GAMESS program for elements up to neon. The PM6 method also involves d-orbitals for elements past neon but the associated integral code has not yet been implemented. Physical constants in the semi-empirical part of the GAMESS source code were updated, to match those in the current version of MOPAC. All semi-empirical methods in GAMESS uses a finite difference scheme for gradient evaluation.

Dispersion correction E(D3)

E(D3) is the third generation dispersion correction developed by Grimme et al. (2010) DFT-D3 and implemented in GAMESS by R Peverati. Unless otherwise noted E(D3) refers to the pair-wise additive dispersion correction as proposed in Grimme et al. (2010). Only the zero-damping version was used, with dispersion order 6 and 8. The fitting parameters are those obtained by Grimme (2012) for PM6. As described by Grimme, the parameter s6 is set to unity, α was set to its default value. s8 and the scaling parameter sr,6 of the atomic cut-off radii used in the dispersion damping function are fitted parameters as in standard DFT-D3 (see Table 1 for parameters). Thus only s8 and sr,6 are optimized by Grimme for PM6-D3H, which is also used for PM6-D3H+. The gradient of the dispersion correction is evaluated numerically, by using a centered finite difference scheme, for three-body calculations, and analytically for two-body calculations.

Table 1 The final parameters for the dispersion and hydrogen bond correction terms of PM6-D3H+.

H+	C N	−0.110	
	C O	−0.120	
D3	α	14.000	
	s 6	1.000	
	s r,6	1.560	
	s 8	1.009	

Hydrogen bond correction E(H+)

E(H+) is a slightly modified version of the third-generation hydrogen bonding correction, H+, by Korth (2010), which is given by: (2) EH+=∑ABCA+CB2rAB2⋅fgeom⋅fbond⋅fdamp

where the sum runs over all hydrogen bonds involving N and O atoms. rAB is the donor–acceptor distance for the given hydrogen bond geometry, with A and B being the two possible acceptor/donor electronegative atoms, either oxygen or nitrogen. CA and CB are adjustable parameters and refer to either CN and CO. CN and CO are re-parametrized as part of this work as described below.

The geometrical correction fgeom is defined as (3) fgeom=cos2θ⋅cos2ϕA⋅cos2ψA⋅cos2ϕB⋅cos2ψB

where θ is the angle defined by atom A, atom B and the hydrogen (see Figs. 1 and 2). The angle ϕ, and torsion angle ψ are both defined by the hydrogen bonding geometry. The angles ϕ are calculated from the difference between the target angle ϕtarget and the present bond angle in the complex ΦX. The target angle ϕtarget is the optimum angle for hydrogen bonds. Target angles are defined in a complicated heuristic fashion, please see the source code posted on GitHub for more details (JensenGroup, 2014b). The torsion angles ψ are defined similarly and calculated as the difference between target dihedral angle and the structural angle Ψ. Where ΨX is the dihedral angle between R1R2X…H, which is used for both the donor and acceptor as seen in Figs. 1 and 2. Here R1 is defined as the Rx closest to the hydrogen.

Figure 1 Illustrating the angles of the H+ model when the hydrogen bond acceptor is sp3 hybridized.

Θ is the angle between atoms A and B. ΦX is the angle between the hydrogen and the R1 atom, H…X-R1, where R1 is the atom closest to the H atom. ΨX is the dihedral angle between R1R2X…H.

Figure 2 Illustrating the angles of the H+ model when the hydrogen bond acceptor is sp2 hybridized.

Θ is the angle between atoms A and B. ΦX is the angle between the hydrogen and the R1 atom, H…X-R1, where R1 is the atom closest to the H atom. ΨX is the dihedral angle between R1R2X…H.

The bond damping function fbond is defined as: (4) fbond=1−11+exp−60⋅rXH/1.2−1

where rXH is the distance between the hydrogen atom and the donor atom, which is defined as the shorter one of the distances rAH and rBH. The damping function fdamp is defined as: (5) fdamp=11+exp−100⋅rAB/2.4−11−11+exp−10⋅rAB/7.0−1

where rAB is the distance between the two electronegative atoms A and B.

The E(H+) implementation differs slightly from the one originally proposed by Korth (2010). Changes were made to avoid problems with optimization of hydrogen bond complexes involving particular configurations, including especially ketone (C=O) groups interacting with amide-like (NR3) groups. In the original implementation, optimization problems can originate from target angle calculation based on the torsion angle of the NR3 group. Target angles are the optimal (text-book) angles for a given H-bond arrangement. H-bond energies are computed based on the deviation of all angular coordinates from their respective target (optimal) angles, see Korth (2011) for a detailed explanation. The target angle would switch during optimization steps as the definition of the torsion angle would switch, and never find a minimum, as the torsion angle is defined as seen in Fig. 1. The model was updated with new target angles for tetragonal NR3 configuration case, and the estimation of target angles for NR3 groups now based on the hydrogen bonding configuration (with a double bond indicating a planar structure).

The analytical gradient is done using internal coordinates from the energy model (angles and distances in Eq. (2)), and an algorithm for converting the gradient to the Cartesian atomic coordinates.

Source code for the H+ module, including gradient code, is available on GitHub (JensenGroup, 2014b). The PM6-D3H+ will be made available in the official version of GAMESS as soon as possible.

Method

All PM6-D3H+ calculations were done with a locally modified version of GAMESS. To benchmark and test our implementation, we performed various calculations on the S22 (Jurečka et al., 2006) and S66 (Řezáč, Riley & Hobza, 2011) complexes from the Benchmark Energy and Geometry Database (BEGDB) (Řezáč et al., 2008). The BEGDB database contains structures and corresponding interaction energies calculated at the MP2/cc-pVTZ and estimated CCSD(T)/CBS level of theory, respectively. We use double displacement for the Hessian calculations (NVIB = 2 in $force group in the GAMESS input file).

Geometry optimizations of the complexes in S22 and S66 were done with a variety of convergence criteria which will be discussed in detail in section ‘Geometry optimization’. Geometry optimizations of Chignolin (PDB: 1UAO) and the Tryptophan-cage (PDB: 1L2Y) using PM6-D3H+ were also carried out. We used the first structure available in each of the downloaded structures. For comparison, we performed two-body Fragment Molecular Orbital (FMO) (Fedorov & Kitaura, 2007) geometry optimizations using RHF/6-31G(d) (Francl et al., 1982; Gordon et al., 1982; Hariharan & Pople, 1973; Nagata et al., 2011) and the D3 dispersion correction (Grimme et al., 2010; Peverati & Baldridge, 2008).

Calculations were performed in either in the gas phase or in bulk solvent using a polarizable continuum to model the solvent (Tomansi, Mennucci & Cammi, 2005). For solvated PM6-D3H+ calculations, we used a recent C-PCM implementation (Steinmann et al., 2013). For the FMO calculations, we used the recent completely analytical RHF/C-PCM gradient (Nagata et al., 2012). All PCM calculations were done using the FIXPVA (Su & Li, 2009) tesselation scheme with 60 tesserae per sphere. All geometry optimizations used a convergence criterion of 5.0 × 10−4 Hartree/Bohr.

All MOPAC calculations were done with MOPAC2012 (Stewart, 2012; Maia et al., 2012). Geometry optimizations were done with the LBFGS optimizer for reasons described in section ‘Geometry optimization’, unless noted otherwise. The COSMO model (Klamt & Schuurmann, 1993) were used to model bulk solvation for the protein calculations.

Timings was carried out on either a 8 core Intel(R) Xeon(R) CPU X5560 @ 2.80 GHz or 24 core AMD Opteron(tm) Processor 6172 @ 2.1 GHz machine.

Results and Discussion

Parameterization of correction terms

Because we use a different dispersion energy function than in the previous DH+ model and make modification to the original hydrogen bonding correction model, it is necessary to determine new optimum values for the CN and CO. The parameters for H+ are parameterized to minimize the root-mean-square deviation (RMSD) between the interaction energies for PM6 with dispersion correction only (PM6-D3) for a subset of structures from the S22 and S66 data sets (1–7 and 1–23, respectively), plus the H+ term and the estimated CCSD(T)/CBS reference interaction energy. The CN and CO parameters are then scanned in ranges from −0.2 to 0.0, around the original optimum. A global optimum was found at CN = −0.11 and CO = −0.12, with a RMSD of 1.11 kcal/mol, as seen in Fig. 3 and Table 2.

Figure 3 Scan of the two parameters for the H+ correction term, nitrogen (CN) and oxygen (CO) in the hydrogen bond dominant complexes of the S22 and S66 noncovalent complexes.

A global optimum was found at CN = −0.11 and CO = −0.12.

This was done using both two and three-body dispersion, but including three-body dispersion did not make any substantial difference in the resulting optimum, and the default was set to two-body for PM6-D3H+, because of the extra computational time associated with three-body. The computational cost becomes a time consuming issue for protein-sized molecules. The final set of parameters for both dispersion and hydrogen bond correction terms can be seen in Table 1.

Interaction energies

Table 2 shows results of PM6, PM6-DH+ and PM6-D3H+ for the full, dispersion and hydrogen bond dominant complexes sets of the S22 and S66 from BEGDB. Root-mean-square deviation (RMSD), mean absolute deviation (MAD) and maximum error span (Max) with respect to the benchmark estimated CCSD(T)/CBS interaction energies are given in kcal/mol. The PM6-D3H+ method was tested using both two and three-body dispersion.

Table 2 Root-mean-square deviation, RMSD; mean absolute deviation, MAD; as well as the maximum error Max, with respect to the estimated CCSD(T)/CBS interaction energies from the S22 and S66 sets are presented.

Hydrogen bond and dispersion subsets are complexes from S22 and S66 with a dominant factor of the interaction energy being hydrogen bond or dispersion interaction. All values are in kcal/mol.

	PM6a,b	DH2b	DH+b	D3H+a,c	D3H+a,d	
	Full set	
RMSD	3.35	0.83	0.80	0.82	0.83	
MAD	2.85	0.58	0.61	0.60	0.61	
Max	7.99	3.53	2.47	2.11	2.09	
	Dispersion subset	
RMSD	3.15	0.49	0.49	0.48	0.54	
MAD	2.79	0.42	0.42	0.36	0.39	
Max	7.29	0.92	0.92	1.11	1.43	
	Hydrogen bond subset	
RMSD	4.29	1.05	0.98	1.11	1.11	
MAD	3.65	0.70	0.80	0.92	0.91	
Max	7.99	3.53	2.10	1.85	1.84	
Notes.

a The calculations have been done using the GAMESS software.

b The calculations have been done using the MOPAC software.

c The calculation has been done using two-body dispersion.

d The calculation has been done using three-body dispersion.

Overall, the accuracy of PM6-D3H+ is very similar to PM6-DH2 and PM6-DH+, with RMSD and MAD values within 0.02 kcal/mol of one another. The main difference is that the maximum error for PM6-D3H+ is 1.42 and 0.36 kcal/mol smaller than for PM6-DH2 and PM6-DH+, respectively. The maximum error for PM6-DH2, -DH+, and -D3H+ were observed for the S66-19, S66-60, and S66-65 dimer, respectively and, in general, we did not notice any particular dimer that resulted in unusually large errors for all three corrections. All interaction energies can be found in Supplemental Information. The differences in RMSD and MAD between methods are slightly larger (up to 0.13 kcal/mol) for subsets where dispersion and hydrogen-bonding dominate. Including three-body dispersion correction had no substantial effect on accuracy, but might play a role for large systems.

Next, we test PM6-DH3+ on two sets of molecules not in the training set. Table 3 lists computed interaction energies for formamide dimer, pentamer–monomer, and trimer–trimer (Fig. 4) computed with various methods. Compared to MP2/TZVP PM6-DH2 performs best for this particular system, while PM6-DH+ and PM6-D3H+ appear to perform roughly similarly, with mean absolute deviations (MAD) of 0.8 and 1.3 kcal/mol, respectively. However, it is interesting to note that the decrease in interaction energy on going from the dimer to the pentamer–monomer predicted by PM6-DH+ (3.6 kcal/mol) is somewhat lower than that predicted by other methods corrections and MP2/TZV (4.1–4.6 kcal/mol). This decrease comes primarily from cooperative polarization effects that are accounted for by the underlying PM6 method, and PM6, PM6-DH2, and PM6-D3H+ all predict similar decreases. It is not clear why the DH+ terms leads to an underestimation of the cooperative effect.

Table 3 Hydrogen bond interaction energies, with various methods, from formamide dimer, pentamer–monomer, and trimer–trimer, as well as MP2/TZVP reference data.

All values are in kcal/mol.

	PM6a,b	DH2b	DH+b	D3H+a	MP2/TZVPc	
Dmer	−5.36	−6.71	−7.81	−8.12	−6.65	
Pentamer–monomer	−7.17	−8.82	−9.56	−10.06	−8.66	
Trimer–trimer	−9.27	−11.33	−11.45	−12.23	−11.26	
Notes.

a The calculations have been done using the GAMESS software.

b The calculations have been done using the MOPAC software.

c From Kobko et al. (2001) and Kobko & Dannenberg (2003).

Figure 4 Illustrating the formamide trimer–trimer (A), hexamer (B) and pentamer–monomer (C).

Table 4 contains RMSD, MAD, mean-deviation (MD) and maximum deviation relative to estimated CCSD(T)/CBS//MP2/pVTZ interaction energies computed for 12 hydrogen bonded base pair complexes (Table S1) from the JSCH-2005 (Jurečka et al., 2006) set from BEGDB. The 12 complexes represent all the complexes in the JSCH-2005 set with hydrogen bonds involving N and O atoms and for which interaction energies have been computed at a level similar to that used in the parameterization of PM6-D3H+ [i.e., CCSD(T)/CBS// MP2/pVTZ]. For this set all three corrections offer very significant increases in accuracy (e.g., a ca 8 kcal/mol decrease in the MAD) compared to PM6. As for the training set (Table 2) the accuracy of PM6-DH2, PM6-DH+, and PM6-D3H+ are very similar, with MADs between 0.7 and 1.1 kcal/mol.

Table 4 Root-mean-square deviation, RMSD; mean absolute deviation, MAD; mean deviation, MD; as well as the maximum error, Max, with respect to the estimated CCSD(T)/CBS interaction energies from selected complexes from JSCH-2005 dataset.

Method	RMSD	MAD	MD	Max	
PM6a,b	8.24	7.98	7.98	10.71	
PM6-DH2b	1.45	1.09	0.21	3.97	
PM6-DH+b	0.94	0.69	0.46	1.90	
PM6-D3H+a	1.18	0.95	0.37	2.45	
Notes.

a The calculations have been done using the GAMESS software.

b The calculations have been done using the MOPAC software.

Geometry optimization

All structures from the S22 and S66 data sets were optimized with PM6, and PM6-DH+ using MOPAC or PM6 and PM6-D3H+ using GAMESS to test how well the methods reproduce the reference MP2/cc-pVTZ geometries and to compare the optimization algorithms in GAMESS and MOPAC.

For the GAMESS optimizations we used the default (quasi Newton–Raphson) geometry optimizer and defined convergence as having a maximum gradient component less than 5 × 10−4 Hartree/Bohr and an RMS gradient less than 5/3 × 10−4 Hartree/Bohr. These convergence criteria are five times higher than the default and are chosen because we have found that for large systems these criteria can lead to significantly faster convergence without affecting the structure or final energy significantly. See supporting information for GAMESS examples of input files. For complex 58 in the S66 set it was necessary to re-compute the Hessian every 20 steps to obtain convergence and in the case of complex 22, 51, and 58 it was necessary to skip the projection of translational and rotational degrees of freedom from the gradient to obtain convergence, which was done by settings the keyword PROJCT=.F. in $Force. For 11 of the complexes (see Table S1) it was necessary to decrease convergence criterion to 10−4 Hartree/Bohr in order to remove imaginary frequencies. In the case of complex 4 and 5 from S22 PM6-D3H+ predicted that the minimum has C1 symmetry rather than Cs as predicted by MP2, and a deviation in the planarity structure of 0.1 Å was needed (added to the first atom). This is not the case for PM6 and thus a result of the D3H+ energy correction.

For the MOPAC optimization we used the LBFGS geometry optimizer because we found that this is the only optimization algorithm that can be practically applied to optimization of large systems. Using eigenvector following leads to termination of the geometry optimization and the following error message: “TRUST RADIUS NOW LESS THAN 0.00010 OPTIMIZATION TERMINATING”. Based on the output the convergence criterion for the LFBGS optimizer appears to be a change heat of formation of less than ca 0.1 kcal/mol during several consecutive optimization steps. For PM6, this convergence test was not passed after ca 200 geometry optimization steps for complex 10 and 17 from the S22 set and 29, 53, and 54 from the S66 set. For PM6-DH+, this convergence failed after ca 140 geometry optimization steps for complex 11 from the S22 set and 53, 54 and 60 from the S66 set. In all these cases MOPAC terminates the geometry optimization after the mentioned number of steps with the message: “A FAILURE HAS OCCURRED”.

The results are summarized in Table 5. The average RMSD between the MP2/cc-pVTZ and semi-empirical structures are below 0.28 Å for all methods and a factor of two lower for the GAMESS optimizations. The RMSD was calculated using the Kabsch algorithm (Kabsch, 1976), for all the atoms, including hydrogens. For the hydrogen bonding subset RMSD was calculated for the hydrogen bond lengths, which are much lower with GAMESS, and with PM6-D3H+ being the lowest with a RMSD of 0.08 Å. The GAMESS optimizations converge, on average, in 30 steps, while the MOPAC optimization takes 10 times more steps.

Table 5 Geometry optimization of equilibrium conformations of the S22 and S66 datasets in gas phase.

Root-mean-square-deviation was calculated between the optimized structures and the original structure from S22 and S66, as well as the hydrogen bond lengths. The average number of steps (N¯S), average of the final root-mean-squared gradient (RMS) in Hartree/Bohr, and average number of imaginary frequencies (N¯i) was noted for the different methods.

	Avg. RMSD (Å)	HB RMSD (Å)	NS¯	Avg. Gradient RMS	N¯i (max)	
PM6a	0.11	0.13	30	1.0 × 10−4	0.02 (1)	
PM6-D3H+a	0.12	0.08	31	1.0 × 10−4	0.07 (1)	
PM6b,c	0.28	0.24	229	1.4 × 10−3	0.71 (6)	
PM6-DH+b,d	0.21	0.24	376	2.3 × 10−3	0.79 (9)	
Notes.

a The calculations have been done using the GAMESS software.

b The calculations have been done using the MOPAC software.

c Averages computed without complexes 10 and 17 from S22 and 29, 53 and 54 from S66, as they did not converge.

d Averages computed without complexes 11 from S22 and 53, 54, 60 and 63 from S66, as they did not converge.

Furthermore, MOPAC optimized geometries tend to have a significantly larger RMS gradient, compared to GAMESS. This leads to significantly more imaginary frequencies in a subsequent vibrational analyses compared to those obtained with GAMESS. In the case of MOPAC 54, 17, 15, and 2 geometries result in 0, 1, 2, and ≥3 imaginary frequencies, while the corresponding numbers for GAMESS are 82, 6, 0, and 0 (Tables S2–S4). Using the (default) eigenvector following algorithm in MOPAC for comparison results in 60, 19, 5, and 4 geometries with 0, 1, 2, and ≥3 imaginary frequencies, respectively, with complexes 1 and 3 from S22 and 1 and 20 from S66 failing the optimization.

For four of the six cases where a GAMESS optimization leads to a structure with a single imaginary frequency a convergence criterion of 10−4 Hartree/Bohr is used, but lowering the convergence criterion further does not remove the imaginary frequencies. In the sixth case, complex 16 in the S66 set (water hydrogen bonded to an amide group— Fig. 5), the optimization stalls, when setting convergence criterion to 10−4 Hartree/Bohr, with the maximum gradient oscillating between 3 × 10−4 and 2 × 10−4 Hartree/Bohr. This is due to the dihedral angle ψ (Eq. (3)) which is defined as R1R2X…H (cf. Figs. 1 and 2), where R1 is defined as the atom closest to the H atom. In the case of the amide–water hydrogen bond, R1 and R2 are the two water H atoms, which are approximately equidistant from the amide proton. The oscillation in the maximum gradient is caused by the oscillation between two different definitions of ψ, which has an effect on the gradient direction. The normal mode associated with the imaginary frequency for the structure converged with a convergence criterion of 5 × 10−4 corresponds to a motion between these two structures, so this is likely the explanation for the imaginary frequency. Similarly, in the case of the complex 1 in the S22 set (ammonia dimer), we believe the imaginary frequency is due to highly symmetric hydrogen configuration, with switching torsion angles (atomic definition of ψ). Since this only affects structures with highly symmetric hydrogen bonds it is unlikely to cause problems in most applications. We note that the PM6-DH+ method has the same problem.

Figure 5 Hydrogen bond configuration of complex 16 of the S66 set.

This figure was made with Jmol.

In the remaining four cases where a GAMESS optimization leads to a structure with an imaginary frequency the cause is most likely an extremely flat potential energy surface for the corresponding degrees of freedom: all imaginary frequencies are <31i cm−1. Similarly, the lowest real frequencies for these five cases are all <40 cm−1.

In summary, the PM6-D3H+ method as implemented in GAMESS offers an attractive alternative to PM6-DH+ in MOPAC in cases where the default geometry optimizer fails to find a converged structure and the LBFGS optimizer must be used and a vibrational analysis is needed e.g., when computing vibrational free energies.

Application to protein structure refinement

In this section we test the applicability of the PM6-D3H+ method, combined with the PCM for bulk solvation as implemented in GAMESS, to geometry optimization of large systems such as proteins and compare to corresponding calculations performed using MOPAC.

We optimize the proteins Chignolin (1UAO) and Trp-Cage (1L2Y), which are two small proteins with 138 and 304 atoms, respectively. We optimize the structures using PM6-DH+ in MOPAC (Stewart, 2012), and PM6-D3H+ in GAMESS (Schmidt et al., 1993), with and without implicit solvent models. The optimized semi-empirical structures are compared to the reference structure optimized at the RHF/6-31G(d) level of theory using dispersion correction (DFTD3) and two-body Fragment Molecular Method (FMO2). Previous calculations by Nagata et al. (2012) have shown that this level of theory yields protein structures in good agreement with corresponding MP2 calculations. Optimized reference structures are available on GitHub (JensenGroup, 2014a).

The results are summarized in Table 6. The RMSD values are about 1 Å in the gas phase for both methods, with PM6-DH+ being slightly smaller. The RMSD values for the structures in solution are slightly larger compared to the corresponding gas phase values for PM6-DH+, and slightly smaller for PM6-D3H+. The structural overlap between PM6-D3H+/PCM optimization and the reference structure can been seen in Figs. 6 and 7. For Trp-cage both methods converge in about half the number of steps compared to gas phase. MOPAC requires significantly more optimization steps than GAMESS to converge, but the overall time for optimization of the structures is by far faster than GAMESS. The difference in CPU time per geometry optimization step is significantly larger for optimization in bulk solvent, which indicates that it is the difference in the COSMO and PCM interfaces that differ most in terms of CPU requirements. Despite being significantly slower than PM6-DH+/COSMO, the PM6-D3H+/PCM implementation in GAMESS is sufficiently fast to make geometry optimizations of small proteins feasible.

Table 6 Optimized proteins Chignolin with 138 atoms and Trp-Cage with 304 atoms, in gasphase and implicit solvent, using PM6-DH+ and PM6-D3H+ with COSMO and PCM respectively for solvent polarization.

RMSD (in Å) are calculated with reference to the protein structures optimized at FMO2-RHF-D3/6-31G(d) level of theory and FMO2-RHF-D3/6-31G(d)/PCM level for solvent effects. Time in hours and number of optimization steps were noted. Calculations was run on a single core.

		PM6-DH+b	PM6-D3H+a	
System	PDB	RMSD (Å)	Time (h)	Steps	Ni	RMSD (Å)	Time (h)	Steps	Ni d	
Chignolin	1UAO	0.90	0.1	739	4	0.98	0.2	204	0 (0)	
Trp-Cage	1L2Y	1.89	1.1	1774	2	1.61	5.4	481	2 (0)	
Chignolinc	1UAO	1.14	0.1	941	5	0.56	0.6	128	3 (0)	
Trp-Cagec	1L2Y	1.23	0.6	882	12	0.83	5.2	174	2 (0)	
Notes.

a The calculations have been done using the GAMESS software.

b The calculations have been done using the MOPAC software.

c Calculations was done using implicit solvent models. PCM for GAMESS, COSMO for MOPAC.

d Number of imaginary frequencies for OPTTOL = 5 × 10−4 (1 × 10−4) aus.

Figure 6 Trp-cage (1L2Y) optimized with FMO2-RHF-D3/6-31G(d)/PCM (black), compared to (A) PM6-D3H+/PCM (blue) and (B) PM6-DH+/COSMO (green).

This figure was made with PyMOL Schrodinger.

Figure 7 Chignolin (1UOA) optimized with FMO2-RHF-D3/6-31G(d)/PCM (black), compared to (A) PM6-D3H+/PCM (blue) and (B) PM6-DH+/COSMO (green).

This figure was made with PyMOL Schrodinger.

The number of imaginary frequencies computed for the optimized protein geometries (Ni) are listed in Table 6. Again, the GAMESS optimization leads to significantly fewer imaginary frequencies: 3 and 3 using PM6-D3H+/PCM implemented in GAMESS, compared to 5 and 12 for Chignolin and Trp-cage using PM6-DH+/COSMO implemented in MOPAC. In the case of GAMESS the number of imaginary frequencies can be reduced to 0 for both proteins by decreasing the geometry optimization criterion (OPTTOL) to 1 × 10−4 aus. This required 205 and 298 additional optimization steps for Chignolin and Trp-cage, respectively.

The relative speedup from running in parallel in solvent is shown on Fig. 8, where no improvement is observed beyond 8 cores for all methods. The timings were done on 24 core AMD Opteron(tm) Processor 6172 @ 2.1 GHz machine for GAMESS and 8 core Intel(R) Xeon(R) CPU X5560 @ 2.80 GHz for MOPAC, because we were unable to get MOPAC running on the AMD ones. Using the dispersion correction and hydrogen bond correction on the PM6 method in GAMESS reduces the relative speedup from 4 to about 2. The correction terms to the PM6 energy only runs in serial, and a modest speedup could be gained by parallelising them. Here we note that the poor scaling of run times with regards to the number of CPUs used is an inherent problem for semi-empirical since the matrix diagonalization in the SCF procedure cannot be efficiently parallelized (Maia et al., 2012).

Figure 8 Speedup by using multiple cores with PCM enabled for single point energy and gradient evaluation of the proteins Trp-Cage (1L2Y) with 304 atoms and Chignolin (1UAO) with 138 atoms, using (A) PM6 and (B) PM6-D3H+ in GAMESS and (C) PM6 and (D) PM6-DH+ in MOPAC.

The evaluation was done using implicit solvent models COSMO and PCM for respectively MOPAC and GAMESS. a The calculations have been done using the GAMESS software. b The calculations have been done using the MOPAC software.

Conclusions

Recent studies by Gilson (Muddana & Gilson, 2012) and Grimme (2012) and co-workers have used dispersion and hydrogen bonded corrected PM6 to compute the vibrational free energy contribution to the standard binding free energy for host–guest systems. However, computing this vibrational free energy contribution can be complicated by the presence of one or more imaginary frequencies in the vibrational analysis, and these numerical problems can introduce a significant error in the binding free energy.

In this paper we address this problem by developing the PM6-D3H+ method and implementing it in the GAMESS program. The method combines the D3 dispersion correction devloped by Grimme and co-workers with a modified version of the H+ hydrogen bond correction developed by Korth and co-workers. Overall, the accuracy of PM6-D3H+ is very similar to PM6-DH2 and PM6-DH+, with RMSD and MAD values within 0.02 kcal/mol of one another. The main difference is that the maximum error for PM6-D3H+ is 1.42 and 0.36 kcal/mol smaller than for PM6-DH2 and PM6-DH+, respectively.

Geometry optimizations of 88 complexes result in 82, 6, 0, and 0 geometries with 0, 1, 2, and ≥3 imaginary frequencies using PM6-D3H+ implemented in GAMESS, while the corresponding numbers for PM6-DH+ implemented in MOPAC are 54, 17, 15, and 2 (Tables S2–S4). This decrease is mainly due to differences in geometry optimization algorithms and convergence criteria.

Furthermore, the numerical stability of the method could be increased by changing the definition of some of the dihedral angles used in the hydrogen bond correction term. However, this appears only to be an issue for very symmetric systems which is unlikely to occur in large heterogenous systems such as proteins.

The PM6-D3H+ method as implemented in GAMESS offers an attractive alternative to PM6-DH+ in MOPAC in cases where the LBFGS optimizer must be used and a vibrational analysis is needed, e.g., when computing vibrational free energies.

While the GAMESS implementation is up to 10 times slower for geometry optimizations of proteins in bulk solvent, it is sufficiently fast to make geometry optimizations of small proteins practically feasible.

Supplemental Information

Supplemental Information 1 Supplementary tables

Click here for additional data file.

Supplemental Information 2 Table of interaction energies of s22 and s66

Click here for additional data file.

All authors would like to thank Jimmy (Mr. Mopac) Stewart for providing PM6 source code for the GAMESS implementation.

Additional Information and Declarations

Competing Interests

Author Contributions

Data Deposition

The authors declare there are no competing interests.

Jimmy C. Kromann conceived and designed the experiments, performed the experiments, analyzed the data, wrote the paper, prepared figures and/or tables, reviewed drafts of the paper.

Anders S. Christensen and Casper Steinmann conceived and designed the experiments, performed the experiments, analyzed the data, wrote the paper, reviewed drafts of the paper.

Martin Korth and Jan H. Jensen conceived and designed the experiments, analyzed the data, wrote the paper, reviewed drafts of the paper.

The following information was supplied regarding the deposition of related data:

Reference protein structures is available on Github: https://github.com/jensengroup/optimized-protein-structures.

Source code of the modified method, Github: https://github.com/jensengroup/hydrogen-bond-correction-f3.

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
