# Peer review of "A third-generation dispersion and third-generation hydrogen bonding corrected PM6 method: PM6-D3H+"

_PeerJ, doi:10.7717/peerj.449_

## Round 0.1 · original submission · Minor Revisions

Dear authors

As you can learn both reviewers have their comments to the submitted article and suggested minor revision of it current version. Please concentrate on misleading statements and add some of the suggested and relevant works. Respond also on the critics that you used numerical derivative of the dispersion energy for the gradient, as the analytical form has been available since 2010.

I hope you can send the revised manuscript soon to be published at PeerJ

Jiri Vondrasek

Reviewer 1 ·

Basic reporting

Very good

Experimental design

The work fulfills all requirements. However, this is a methodological computational study with little overlap with biology and I am not sure that it fits in the scope of PeerJ (Biological Sciences, Medical Sciences, and Health Sciences).

Validity of the findings

Some of the conclusions may not be appropriately stated, see below.

Additional comments

This ms presents an implementation of a slightly modified empirical correction to PM6 method, D3H+. The correction is implemented in GAMESS code.

In my opinion this work is well performed, clearly written and suitable for publication in PeerJ. However, there are two points that should be addressed before publication:

1. Misleading statement.
In the abstract, the authors say “The main difference is that the geometry optimizations of 88 complexes result in 82, 6, 0, and 0 geometries with 0, 1, 2, and $\ge$ 3 imaginary frequencies using PM6-D3H+ implemented in GAMESS, while the corresponding numbers for PM6-DH+ implemented in MOPAC are 54, 17, 15, and 2.”

This may imply that the D3H+ correction implemented in GAMESS is in this respect better than the DH+ correction implemented in Mopac. However, this is not the case, because the difference comes from the quality of optimization algorithms and convergence criteria. Later in the text they clarify the results, stating that:

“However, we show that PM6-D3H+ implemented in GAMESS results in vibrational analyses with significantly fewer imaginary frequencies than PM6-DH+ implemented in MOPAC[12, 13], due mainly to differences in geometry optimization algorithms and convergence criteria.”

In my opinion the authors should make it clear in the abstract and also in the conclusions that the reduced number of imaginary frequencies is not due to quality of their modified method, but due to quality of the optimizers and convergence criteria. They should either remove the misleading statement from the abstract or clarify it. The misleading statement should be clarified also in Conclusions.

2. Some relevant works are not cited.
In 2012 another correction to PM6 method has been published (PM6 D3H4), which showed very good results. It would be fair to compare the new PM6 D3H+ parameterization with the previously published results. If this is technically too complicated, the work should be at least cited.


Minor points:

It is not clear what is meant by “pentamer-monomer, and trimer-trimer” in Table 3 and I could not find any explanation in the text.

On page 8, authors write “… on going from the dimer to the pentamer-trimer more than other methods.” but there is no reference to “pentamer-trimer” anywhere in the text. Perhaps a typo?

Another typo on the same page: “to CCSCD(T)/CBS”

Reviewer 2 ·

Basic reporting

No Comments.

Experimental design

No Comments.

Validity of the findings

No Comments.

Additional comments

In the MS 'A third-generation dispersion and third-generation hydrogen corrected PM6 method: PM6-D3H+' the authors
present a new combination of already established methods, namely the emiempirical PM6 method, the DFT-D3 dispersion correction and the H+ hydrogen bonding correction for semiempirical methods. The method is also implemented into the
GAMESS program, and tested for various systems. While the method performs similarly to its predecessors (PM6-DH2 and PM6-DH+), the main improvement is the combination with the geometry optimization algorithm present in GAMESS, which is much more efficient than the one implemented in MOPAC, the program package that features PM6-DH2 and PM6-DH+. On the other hand, the MOPAC-implementation of PM6 (which is the time-consuming step in those calculations) is about 10 times faster than the implementation in GAMESS, which makes MOPAC the overall more efficient choice.

However, it is also stated that the GAMESS implementation results in better convergence of the molecular geometry, which is a very desirable property, expecially when thermochemical data is needed.

Overall, the data presented in the paper, including the newly proposed PM6-D3H+ method, ist certainly worth publishing. However, there are several scientific and formal issues that have to be adressed prior publication:

1.The statement that semi-empirical methods yield interaction energies that rival those computed with Density Functional Theory is clearly too general. While it is certainly clear that this can be the case for some systems (as provided in the references), there are many more examples where this is just not true.

2. In section 3.2, the authors claim that DFT-D3 was implemented in GAMESS by Peverati et al. and give a literature reference from 2008. This is clearly not possbile, as the originial DFT-D3 publication is from 2010.

3. It is not clear why the authors use a numerical derivative of the dispersion energy for the gradient, as the analytical form has been available since 2010 and should be preferred for the sake of efficiency and numerical accuracy.

4. In section 5.2, the second part of the third paragraph is not very clear. The authors not that 'PM6-DH+ underestimates the decrease in interaction energy on going from the dimer to the pentamer-trimer more than other methods'. While there is no pentamer-trimer in table 3, this sentence also does not make any sense if one substitutes pentamer-trimer with trimer-trimer or pentamer-monomer, as the data present in the table do not support this claim.

5. In section 5.3, the authors write that 'the convergence criterion appears to be a change in [...]'. The convergence criterion is of high importance when discussing the convergence behavior of different optimizers, thus a more accurate statement
is in order here.

6. The discussed oscillating behavior that can appear in the H+ correction is quite alarming. While the authors claim that this is only the case for highly symmetric hydrogen bonds and thus 'unlikely to cause problems in most application', it can in principle occur in every interaction with e.g. water or ammonia. As they claim that 'the numerical stability could be increased by changing the definitions of some dihedral angles' in the conclusion, the question remains while this was not done in the course of the work done for this publication.

7. The term CCSD(T)/CBS is used quite often in the MS; as these are clearly estimations of the CCSD(T)/CBS energies in all cases, this should be made clear (e.g. mention it in the text or use 'est. CCSD(T)' in the text).

8. There are several orthograpic mistakes in the paper, which should be adressed. Here is a (probably incomplete) list:

Grimme et al. misses the full stop after 'al' in the abstract
'analyse' should read 'analysis' on page 2
'resent' should read 'recent' on page 5
'criteria' should read 'criterion' on page 5
'an convergence' should read 'a convergence' on page 12
'Figure 5 and 6' should read 'Figures 5 and 6' on page 14

·

Basic reporting

see below

Experimental design

see below

Validity of the findings

see below

Additional comments

This paper reports a useful contribution to computational chemistry software. The new code provides users with the means to energy minimize molecules and supramolecular complexes with the PM6 semi-empirical model, coupled with empirical hydrogen-bonding and dispersion terms, in a more complete manner than was hitherto possible. By allowing the use of a more powerful energy-minimization algorithm, the code reduces the number of negative eigenvalues remaining at local energy minima, and hence facilitates thermochemistry calculations based on normal mode analysis. This capability will be useful for binding energy and other chemically and biomedically relevant applications. This is a solid contribution.

I have one substantial recommendation to improve this paper. Given that a central purpose of the study is to develop a method that gives energy minima of small proteins with fewer negative eigenvalues, it would seem to be of central importance to report the numbers of negative eigenvalues for the energy-minimized structures in Table 6, for both MOPAC and the new method.

Page 15, first line. The text says there is no improvement in speed after 4 cores, but GAMESS does show improvement on going from 4 to 8 cores (Figure 7).

A few suggestions for improving clarity are presented below.

Page 2, line 2: It’s not clear in what respect the PM6xxx energies rival those from DFT. How about “rival in accuracy those computed…”?

Page 2, pgph 3: There is a non sequitur between the first and second sentences. Why would implementing a new variant of PM6xxx test whether the problem of imaginary frequencies could be solved by using a different optimizer? I think the answer becomes reasonably clear on further reading, but it would be very helpful to readers if the text could, at this point, clarify that MOPAC doesn’t have any more advanced optimizers, so the authors decided to reimplement PM6xxx in GAMESS, so that its more powerful optimization methods could be used. Also, this paragraph is written in a rather roundabout manner which makes it hard to figure out exactly how the new method relates to prior ones. A crisp statement that PM6 is as before, D3 is as in Grimme, and H+ is a modified version of the prior H+ (or something like this) would be helpful. By the way, given that the present H+ evidently differs somewhat from the prior one, it may avoid confusion if the authors could give it another name; e.g., H*.

Page 4: It would be helpful if the legends of Figures 1 and 2 could state how the R1 and R2 substituents are distinguished, even though this information is eventually found in the text (page 12). Also, it seems odd that the two figures have identical legends. Given that the figures are different, presumably the legends should also be.

Page 4, middle of the page: the sentence beginning “In the original implementation, optimization problems…” mentions a problem with “target angle estimation”, but it isn’t clear what a “target angle” is, why it is estimated instead of computed, or what the torsion angle of the NR3 group has to do with it. It would be helpful to either clarify this, or else perhaps describe it in more general terms that don’t raise these technical questions.

Page 5, penultimate line: was = were.

Page 6, line 3: parametetrized= parameterized

Page 8, second paragraph: it would be interesting to know which dimers gave the greatest errors. Also, did the same dimers give trouble across all models, or did different models run into trouble for different cases?

Page 8, paragraph 3: it would be good to provide a bit more information on the systems not in the training set. Are these drawn from BEGDB? On what basis were they selected, in any case?

Page 11, paragraph 2: what is meant by “a deviation in the planarity structure of 0.1 Angstrom was needed…”?

Page 11, paragraph 3: what are “trust radius issues”? Also, what does it mean for convergence to fail after 140 steps, for example? Would convergence have succeeded if more steps had been carried out?

Page 11, last paragraph: it would be nice to provide readers with a sense for why MOPAC has so much trouble with these minimizations. Are you saying there is something problematic about the energy model, or would minimization succeed if MOPAC had the LBFGS optimizer?

Page 12, second paragraph line 2: it = is

Page 12, third paragraph: it would be helpful if the authors could shed some light on what kind of cases require the LBFGS optimizer. Is it just that MOPAC, for uncertain reasons, fails to eliminate negative eigenvalues for some systems? If so, then it might be clearer to say these are the systems the authors have in mind; they may then state that LBFGS is helpful in such cases. My point, in part, is that these are not so much systems that require the LBFGS optimizer, as hard-to-optimize systems that require an optimizer better than the one in MOPAC. It is probable that an optimizer other than LBFGS would also work, so characterizing these as systems where LBFGS “must be used” is a bit imprecise.

Page 14, third paragraph: there is a sentence which says MOPAC requires significantly less CPU time and then that it is far faster than GAMESS; this seems repetitive.

Page 14, third paragraph: Based on the data in Table 6, I disagree with the statement that the difference in CPU is larger for optimization in bulk solvent; this holds for chignolin but not for Trp cage.

Page 14, third paragraph, last sentence: either delete “practically, since it seems to say the same thing as “feasible”; or else edit the sentence to say something more precise.

---

## Round 0.2 · accepted · Accept

You answered all the questions of referees and suggested changes to your manuscript were implemented. As a result your paper has been accepted to PeerJ.